# EnsemblePigDet: Ensemble Deep Learning for Accurate Pig Detection

Hanse Ahn [1], Seungwook Son [1], Heegon Kim [1], Sungju Lee [2,*], Yongwha Chung [1,*] and Daihee Park [1]

1 Department of Computer Convergence Software, Korea University, Sejong 30019, Korea; hansahn@korea.ac.kr (H.A.); sso7199@korea.ac.kr (S.S.); khg86@korea.ac.kr (H.K.); dhpark@korea.ac.kr (D.P.)
2 Department of Software, Sangmyung University, Cheonan 31066, Korea
* Correspondence: peacfeel@smu.ac.kr (S.L.); ychungy@korea.ac.kr (Y.C.)

**Abstract:** Automated pig monitoring is important for smart pig farms; thus, several deep-learning-based pig monitoring techniques have been proposed recently. In applying automated pig monitoring techniques to real pig farms, however, practical issues such as detecting pigs from overexposed regions, caused by strong sunlight through a window, should be considered. Another practical issue in applying deep-learning-based techniques to a specific pig monitoring application is the annotation cost for pig data. In this study, we propose a method for managing these two practical issues. Using annotated data obtained from training images without overexposed regions, we first generated augmented data to reduce the effect of overexposure. Then, we trained YOLOv4 with both the annotated and augmented data and combined the test results from two YOLOv4 models in a bounding box level to further improve the detection accuracy. We propose accuracy metrics for pig detection in a closed pig pen to evaluate the accuracy of the detection without box-level annotation. Our experimental results with 216,000 "unseen" test data from overexposed regions in the same pig pen show that the proposed ensemble method can significantly improve the detection accuracy of the baseline YOLOv4, from 79.93% to 94.33%, with additional execution time.

**Keywords:** agriculture IT; computer vision; pig detection; deep learning; data augmentation; model ensemble



## 1. Introduction

The health and well-being of group-housed pigs can be maintained by detecting or managing problems regarding their health and welfare in the early stages [1–5]. The reduction of practical problems (e.g., infectious diseases, hygiene deterioration) with individual pigs is essential, as pigs that roam around in an enclosed pen have a high possibility of being infected by diseases or developing stress [6]. However, in general, the farm workforce is significantly low compared to the number of pigs. For example, the pig farm from which the video monitoring data were obtained had more than 1000 pigs cared for by each worker. It is nearly impossible for a small workforce to manage a huge number of pigs. Therefore, the main objective of this study was to identify the number of pigs in a pig pen and to prevent deaths of individual pigs due to health and welfare problems by detecting irregularities.

Many studies have reported the use of monitoring techniques to solve problems in pig pens [7–30]. It is important to detect individual pigs in each video frame to analyze this type of motion behavior, as object detection is the first process for various vision-based high-level analyses. While many researchers have reported the detection of pigs using typical learning and image processing techniques, the detection accuracy for highly occluded images may not be at an acceptable level. Recently, end-to-end deep learning techniques have been proposed for object detection, and various pig-detection methods based on deep learning results (along with the typical learning and image processing techniques) have been reported [11–30]. YOLOv4 [31] is a recently released detector that

can detect pigs with a good tradeoff between speed and accuracy; hence, we used YOLOv4 as the baseline detector in this study.

However, practical issues such as detecting pigs from overexposed regions, caused by strong sunlight through a window, should be considered when applying automated pig monitoring techniques to real pig farms. Image processing and deep learning techniques can be applied to solve such problems caused by sunlight. For example, images with overexposed regions can be generated using image processing and deep learning techniques. However, the overexposed regions caused by strong sunlight through a window depend on the structure of the pig pen; moreover, each pig moves independently within the overexposed regions. Therefore, it is very difficult to generate training images with overexposed regions for different types of pig pens. In this paper, a method utilizing image processing and deep learning techniques is proposed to improve the accuracy of detection, without generating training data.

After training on a dataset that completely excluded data from overexposed regions, the detection accuracy on data from unseen overexposed regions was measured to validate the ability to detect pigs from various forms of overexposed regions. Specifically, a test was performed on the entire two hours of video (216,000 raw video frames) that included the overexposed region after training on extracted data that excludes frames from 8:00 to 10:30 in the morning (during which time an overexposed region was created due to the effect of sunlight from the window; see Figure 1).

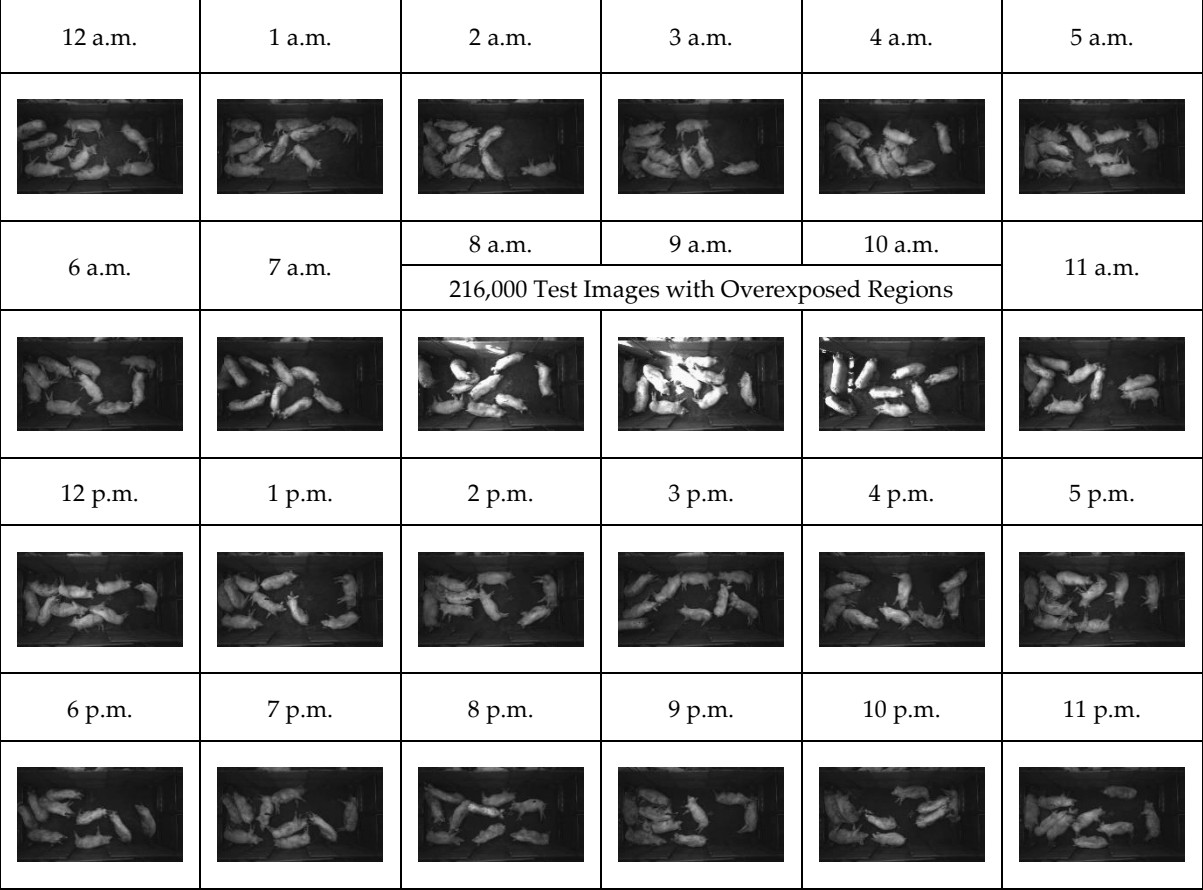

**Figure 1.** Infrared images obtained from a closed pig pen at each hour. From 8:30 a.m. to 10:30 a.m., overexposed regions (with grey pixel values higher than 240) caused by sunlight can be observed. To solve this issue, the training images do not include overexposed regions, and the test images include overexposed regions. Practical accuracy metrics are also proposed for evaluating the detection accuracy with 216,000 test images, without box-level annotation.

Another practical issue in applying deep-learning-based techniques to pig detection is the annotation cost for large-scale training and test data. Traditional metrics, such as average precision (AP)/average recall (AR), which are widely used in COCO [32] and VOC [33], are employed to evaluate the detection accuracy of group-housed pigs. These require box-level annotation for each pig in an image. Because of this annotation cost (which is typically 5 min for one image of group-housed pigs), previous studies on pig detection have reported detection accuracy with a small number of test images. For example, if one image takes about five minutes to annotate, then about 18,000 h (estimated to be 2 years) are taken to annotate 216,000 test images, and 1166 h (estimated to be 1.5 months) are taken to annotate 13,997 key frame images. For large-scale evaluation, accuracy metrics without box-level annotation are needed.

This study proposes a performance testing method and performance metrics that are used to compare the accuracy of the proposed ensemble detectors and baseline detectors in processing all frames of the 216,000 images, without box-level annotations. Hence, the number of test data is reduced by extracting key frames for movements that occur by considering the characteristics of pigs that have been lying down for a long time. In addition, this study proposes an accuracy measurement method that does not require additional annotation cost by modifying CorLoc [34], which is used in weakly supervised object detection (WSOD). The validity of this measurement metric can be verified by comparing the pig detection performance in an enclosed pen on all video frames without annotation with the proposed performance metric. This is possible by visual verification of a small number of 13,997 key frame images. The contributions of the proposed method are summarized as follows:

- For real-time deployment, a deep-learning-based pig detector should handle unseen data. An ensemble-based pig detection method is proposed in this study to improve the detection accuracy in overexposed regions (as an example of unseen data), presumably for the first time. The detection of pigs from such overexposed regions is very challenging because the pixel distribution of such regions caused by strong sunlight through a window is different from that of other regions. Without using training data, including those from overexposed regions, image preprocessing for diversity and a model ensemble with different preprocessed images can robustly detect pigs from overexposed regions.
- Another practical issue in applying deep-learning-based techniques to pig detection is the annotation cost for large-scale data. Experimental results for pig detection with large-scale test data have not yet been reported because the box-level annotation cost for this data is very expensive. Accuracy metrics for pig detection in a closed pig pen are proposed to evaluate the accuracy of detection, without box-level annotation. Presumably, this is the first report of large-scale pig detection in a pig pen with 216,000 test data, without any box-level annotation. It is also indicated that the detection accuracy with 216,000 raw video frames is very similar to that with 13,997 key frames. Thus, reducing the number of test images using key frame extraction is effective in reducing both the evaluation cost (with very large test data) and the inference time (with the model ensemble).

This paper is organized as follows: Section 2 summarizes previous pig detection methods. Section 3 describes the proposed method to efficiently detect pigs using the model ensemble method. Section 4 explains the details of the experimental results, along with the new accuracy metric, and the paper is concluded in Section 5.

## 2. Background

The main objective of this study was to automatically monitor and analyze the behavior of an individual pig for 24 h using computer vision methods. Many studies have analyzed the behavior of group-housed pigs. For example, research on pig behavior analysis [7,8], weight measurement [9], environment control [10], pig detection [11–15], tracking [16–18], and segmentation [19] using image processing have been reported. In

addition, pig detection [20–24], behavior analysis [25], and posture detection [27,28] using deep learning have also been reported. Seo, J. et al. [29] proposed a method to solve the problem of process time and accuracy when detecting individual pigs in a resource-limited embedded environment. Considering the high complexity of deep learning for individual pig detection, parallel pipeline processing and filter clustering methods were used to solve the problem of accuracy and process time, allowing the process in an embedded system. The study applied an image processing method to recover the dropped accuracy using filter clustering. Cowton, J. et al. [30] proposed a method of tracking individual pigs in various environment by combining deep learning detection and a tracking algorithm. More specifically, Faster RCNN and DeepSORT were used after organizing a dataset of 1646 images for individual tracking in a low-light environment. However, since the detection method for 24 h individual pig monitoring is dependent on individual pig detection, there is a need for a method to improve individual pig detection accuracy using deep learning that considers special condition of an overexposed region in a pig pen environment. In other words, special conditions such as overexposed regions of the pig room caused by sunlight have to be considered to automatically monitor and analyze the behavior of an individual pig for 24 h. Our proposed study focused on deep learning technology that improves individual pig detection accuracy. This study proposes a method to improve accuracy by combining two models ($CLAHES_{FB}$ and $CLAHE_{ET}$) that considers difficult overexposed regions in pig detection.

Infrared input images were used to detect individual pigs during the day and night. Figure 2 shows the results of deep learning (i.e., YOLOv4 [31]) during the day, night, and daytime with sunlight. During the day, pigs tend to move more actively than at night, as shown in Figure 2a. In contrast, the pigs tend to sleep at night, leading to less movement than during the day, as shown in Figure 2b. During both day and nighttime environments, deep learning methods can provide high detection accuracy for detecting individual pigs. Note that YOLOv4 [31] is a recently released detector that can detect pigs with a good tradeoff between speed and accuracy. However, in environments with strong sunlight, as shown in Figure 2c, the detection accuracy can be degraded by an overexposed environment.

As mentioned in the introduction, it is very difficult to generate training images from overexposed regions for different types of pig pens. Instead of this approach, the overexposed regions with gray pixel values higher than 240 were considered as occlusion (i.e., invalid data), and a method is proposed to improve the detection accuracy with valid data only (i.e., gray pixel values lower than 240). From the training images without gray pixel values higher than 240, two different parameters for image preprocessing were derived to maximize the diversity of the given images. Then, a model ensemble method was developed for combining the test results from the two YOLOv4 models in a bounding box level to further improve the detection accuracy.

Previous studies on pig detection have reported detection accuracy with a small number of test images because of annotation cost. Table 1 shows the pig detection results with 100–1792 test images during the last 10 years.

For large-scale evaluation, accuracy metrics without box-level annotation are required. As the number of pigs in a closed pig pen is known, two accuracy metrics are proposed based on the number of pigs in the pen and the observed number of pigs in images. The effectiveness of the accuracy metrics was also evaluated by defining "key frames" (frames that are considered to have captured meaningful movements of pigs) and "hard frames" (key frames that have located pigs in overexposed regions, that is, frames where accurate detection of pigs is difficult due to overexposed regions). The accuracy metrics were then compared with all frames, key frames, and hard frames, without expensive box-level annotation costs.

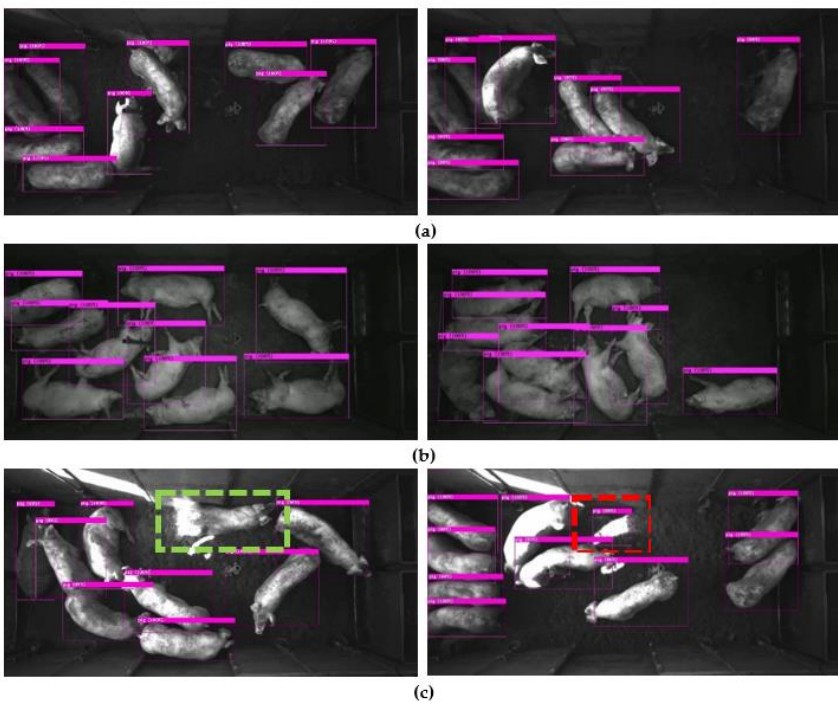

**Figure 2.** Individual pig detection results (infrared-based). (**a**) Daytime (12 p.m. and 3 p.m.); (**b**) nighttime (0 a.m. and 3 a.m.); (**c**) daytime with sunlight (9 a.m. and 10 a.m.). Missing (false negative) and false pigs (i.e., false positive) can be seen.

**Table 1.** Some of the recent results for group-housed pig detection (published during 2011–2020).

| Management of Overexposed Region | Data Size | No. of Pigs in a Pen | No. of Test Images | Detection Technique | Reference |
|---|---|---|---|---|---|
| No | 640 × 480 | 22 | 270 | Image Processing | [8] |
| | 720 × 540 | 12 | 500 | Image Processing | [11] |
| | 1280 × 720 | 7–13 | Not Specified | Image Processing | [12] |
| | 640 × 480 | 22, 23 | Not Specified | Image Processing | [13] |
| | 1440 × 1440 | 9 | 200 | Image Processing | [14] |
| | 1024 × 768 | 4 | 100 | Image Processing | [15] |
| | 720 × 576 | 10 | Not Specified | Image Processing | [16] |
| | Not Specified | 3 | Not Specified | Image Processing | [17] |
| | 512 × 424 | 19 | Not Specified | Image Processing | [18] |
| | Not Specified | 2~12 | 330 | Image Processing | [19] |
| | 2560 × 1440 | 4 | 100 | Deep Learning | [20] |
| | 960 × 720 | ~30 | 500 | Image Processing | [21] |
| | 1920 × 1080 | Not Specified | 400 | Deep Learning | [22] |
| | 64 × 64 | 6 | 500 | Deep Learning | [23] |
| | 1280 × 720 | ~79 | 160 | Deep Learning | [24] |
| | 1280 × 720 | 9 | 1000 | Image Processing + Deep Learning | [25] |
| | 1280 × 720 | ~32 | 400 | Deep Learning | [26] |
| | 1280 × 800 | 13 | 226 | Deep Learning | [27] |
| | 720 × 480 | 2 | 1792 | Deep Learning | [28] |
| | 1280 × 720 | 9 | 1000 | Image Processing + Deep Learning | [29] |
| | 1920 × 1080 | 20 | 828 | Deep Learning | [30] |
| Yes | 1280 × 720 | 9 | 216,000 (13,997 Key Frames) (4193 Hard Frames) | Image Processing + Deep Learning (+ Accuracy Metrics) | Proposed Method |

## 3. Proposed Method

This study proposes a solution to the problem of decreasing accuracy caused by overexposed regions with strong sunlight by applying image preprocessing to images that are not exposed to strong sunlight. In addition, a model ensemble method of combining detection results using the detection boxes of two models to improve detection accuracy is proposed, along with a method for extracting key frames that have effective movement in a test video and new accuracy metrics that measure accuracy, without additional box-level annotation cost. The entire structure of the proposed method is shown in Figure 3.

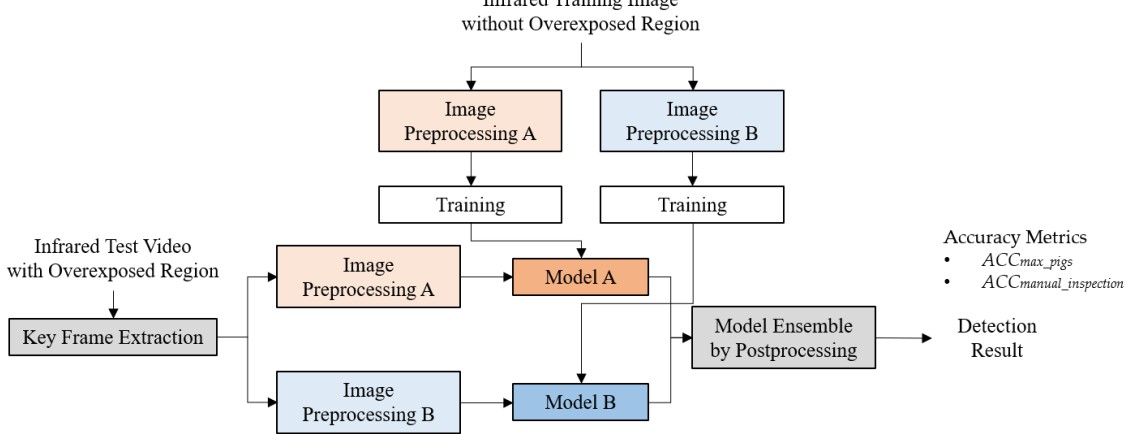

**Figure 3.** Overview of the proposed method EnsemblePigDet.

### 3.1. Image Preprocessing

In this study, detection accuracy is considered to be decreased due to occlusion when the images exposed to strong sunlight have pixel values higher than 240. To solve this problem, image preprocessing on training data that does not include overexposed regions is proposed. It considers overexposed regions with pixel values higher than 240 as invalid regions and increases accuracy by improving image quality through image preprocessing in the remaining regions. This study proposes image preprocessing methods that include contrast limited adaptive histogram (CLAHE) [35], Gaussian filter [36], and sharpening filter [36] and divides them into two sections.

CLAHE divides the input image into small blocks of uniform sizes and smoothens the histogram for each block. The main parameter, TilesGridSize, determines the block sizes to be divided, and ClipLimit is a threshold value that is needed for the histogram smoothing process. This is used to redistribute the pixels that exceed the threshold value and equalize the histogram. As TilesGridSize decreases, it has the effect of increasing local contrast between the object and background, and as it increases, the overall feature strengthens, which emphasizes the object's texture. For this study, averages of individual pixel values in the ground truth box region and ground truth excluded region were acquired to find the effective parameter combination of the first model. This emphasizes the effect of the contrast between the object and background. Among the combinations, the parameter combination that had the largest difference between the averages was chosen. Following this, the entropy value, which is one of the metrics that shows the amount of information in an image, was calculated to find the parameter combination that highlights the texture intensity of an image. Table 2 shows the difference in average pixel value of the object and background (difference) and entropy value (entropy) for each parameter combination.

**Table 2.** Comparison of difference and entropy values for each combination of CLAHE parameters.

| TilesGridSize | ClipLimit | Difference | Entropy |
|:---:|:---:|:---:|:---:|
| (2,2) | 0.2 | 51.14 | 5.48 |
| (2,2) | 0.4 | 51.30 | 5.49 |
| (2,2) | 0.6 $CLAHE_{SFB}$ | 51.31 | 5.49 |
| (2,2) | 0.8 | 51.30 | 5.49 |
| (2,2) | 1.0 | 51.30 | 5.49 |
| (4,4) | 0.2 | 21.44 | 5.57 |
| (4,4) | 0.4 | 17.75 | 5.60 |
| (4,4) | 0.6 | 18.13 | 5.60 |
| (4,4) | 0.8 | 18.15 | 5.60 |
| (4,4) | 1.0 | 18.15 | 5.60 |
| (8,8) | 0.2 | 17.91 | 5.54 |
| (8,8) | 0.4 | 9.27 | 5.60 |
| (8,8) | 0.6 | 7.33 | 5.61 |
| (8,8) | 0.8 | 7.45 | 5.61 |
| (8,8) | 1.0 $CLAHE_{ET}$ | 7.46 | 5.62 |
| (16,16) | 0.2 | 13.41 | 5.45 |
| (16,16) | 0.4 | 4.63 | 5.54 |
| (16,16) | 0.6 | 0.78 | 5.57 |
| (16,16) | 0.8 | 0.69 | 5.58 |
| (16,16) | 1.0 | 1.24 | 5.59 |
| (32,32) | 0.2 | 17.13 | 5.37 |
| (32,32) | 0.4 | 7.82 | 5.47 |
| (32,32) | 0.6 | 1.77 | 5.51 |
| (32,32) | 0.8 | 1.81 | 5.54 |
| (32,32) | 1.0 | 4.19 | 5.55 |
| (64,64) | 0.2 | 18.68 | 5.25 |
| (64,64) | 0.4 | 11.54 | 5.39 |
| (64,64) | 0.6 | 6.15 | 5.44 |
| (64,64) | 0.8 | 1.01 | 5.48 |
| (64,64) | 1.0 | 2.85 | 5.50 |

In this study, $CLAHE_{SFB}$ (i.e., the separation of foreground and background) was assigned as the parameter amongst the CLAHE parameter combinations when using the results of CLAHE image processing for data augmentation. This combination is deemed to separate the foreground from the background most effectively because the difference between the foreground and background pixel average is the highest. On the other hand, $CLAHE_{ET}$ (i.e., enhancement texture) is assigned as a parameter that is deemed to emphasize the texture of an image, as the entropy value is the largest (see Table 2). A Gaussian filter was then applied to the result of $CLAHE_{SFB}$ to emphasize low-frequency features, whereas a sharpening filter was applied to the result of $CLAHE_{ET}$ to emphasize high-frequency features. Finally, "model A" was created by training YOLOv4 using training data applied with image preprocessing A. In contrast, "model B" was created by training YOLOv4 using training data applied with image preprocessing B.

### 3.2. Model Ensemble with Two Models

Although the model ensemble can increase accuracy, it also increases the execution time. Therefore, this study proposes a model ensemble method using information from the detection box of two models at the post-processing level to increase detection accuracy. Initially, a union (AB_box) of the detection box set is acquired from model A and model B using two sets, A_box and B_box, which are sets of detection boxes from models A and B, respectively. After boxes with a lower confidence score for each set of boxes in A_box, B_box, and AB_box are removed, non-maximum suppression (NMS) is performed for each set. The thresholds for confidence and NMS are set differently with a set of model boxes,

A_box and B_box, and a union set of model box AB_box, for diversity. For example, the boxes from set A_box and B_box can be removed aggressively (in Figure 4, Confidence1 and Threshold1 are set to high and low, respectively), whereas the boxes from set AB_box can be removed conservatively (in Figure 4, Confidence2 and Threshold2 are set to low and high, respectively). The effects of diverse combinations of thresholds are explained in Section 4. After the NMS process, the NMS results of A_box and AB_box are combined as A/AB_box using the box merging algorithm (explained later), and the NMS results of B_box and AB_box are combined as B/AB_box. Finally, the detection boxes of A/AB_box and B/AB_box are combined as final boxes, using the same box merging algorithm. Hence, this method merges the detection boxes produced from two models in two steps, with a union set of the model box, to maximize the effect of the model ensemble. The overall structure is shown in Figure 4.

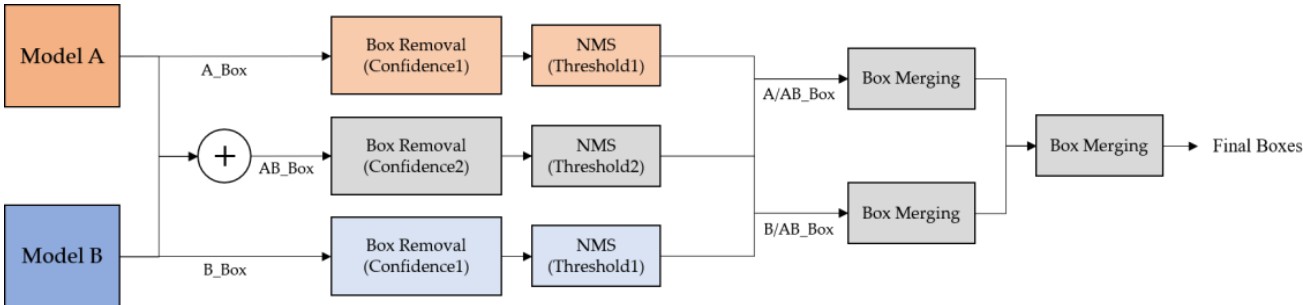

**Figure 4.** Structure of model ensemble.

The box merging algorithm proposed in this study assumes that the number of pigs in a closed pig pen (i.e., no_pigs) is known. For continuous pig monitoring applications, note that the number of pigs in a pen does not generally change for a long time (i.e., one month) except during some events (such as the death of a pig). Furthermore, the proposed algorithm can be modified by introducing thresholds to select the final boxes if the reasonable assumption cannot be satisfied. Another point to note is that the number of pigs (by manual inspection of each video frame) can be less than no_pigs because of possible occlusion. Therefore, the notation max_pigs was used in this study to represent the known number of pigs in a pen.

Each video frame is deemed to be correct when the number of bounding boxes detected by one of the two models matches the number of pigs that can be detected (i.e., no_pigs). If neither of the two models matches, a box that is produced from a trained model that has the largest intersection over union (IOU) value with a box of another model is considered to be a matching box. Subsequently, if this IOU value is higher than iou_thresh, which is a value deemed for two boxes to be the same box (set to 0.7 in this work), it is considered that the matching box has correctly detected the pig. Finally, boxes that have the highest confidence value compared to the remaining boxes that are not chosen as correct boxes from each model are considered until the total number of correct boxes is equal to no_pigs. Hence, this is a merging method in which the detection boxes are produced from two models using the number of pigs that can be detected within a pen at the postprocessing level. The proposed box-merging algorithm is summarized as Algorithm 1.

---

**Algorithm 1.** Box merging algorithm.

---

Input: First boxes *first_box*, Second boxes *second_box*, *no_pigs*
Output: Merged boxes *result_box*
Initialize: Input boxes in first set *first_box*
    Input boxes in second set *second_box*
if (confidence sum of *first_box* $\geq$ confidence sum of *second_box* & size of *first_box* = *no_pigs*) **do**
  return *first_box* as *result_box*
else if (size of *second_box* = *no_pigs*) **do**
  return *second_box* as *result_box*
else **do**
  matched_boxes = 0
  sort *first_box* and *second_box* in descending order of confidence value
  for i = 1 to size of *first_box* **do**
   for j = 1 to size of *second_box* **do**
   max_iou = largest IOU of *first_box*[i] and *second_box*[j]
   if max_iou > *iou_thresh* **do**
    matched_boxes++
    *result_box*[matched_boxes] = *first_box*[i]
   if matched_boxes < *no_pigs* **do**
    for k = matched_boxes + 1 to *no_pigs* **do**
     add remaining *first_box* or *second_box* into *result_box*
  return *result_box*

---

### 3.3. Key Frame Extraction and Accuracy Metrics

To measure the detection accuracy of the test dataset, an expensive "box-level" annotation to create ground truth is usually required. Therefore, a method to measure the detection accuracy at an inexpensive cost is necessary for continuous pig monitoring applications. Raw video frames have a lot of redundant information for pig monitoring because pigs sleep frequently for a long time. In this study, key frames that show significant changes in movement (hence, pigs that show movement) were chosen to reduce the test dataset.

After reducing the test dataset through key frame extraction, CorLoc [34] used in weakly supervised object detection (WSOD) was modified to measure the detection accuracy of key frames, under the assumption of knowing the number of pigs to be detected (max_pigs). This study proposes a method to measure the detection accuracy without the box-level annotation cost.

To extract key frames, the number of pigs that show movement in a current video frame is estimated by comparing the previous and current frames. YOLOv4 is then applied to calculate average size of bounding boxes. Note that YOLOv4 is applied only for a sample frame from training dataset, not input video data. The pixel difference between the previous and current frames is computed to decide whether a pig is moving. Finally, the number of pigs that are moving is estimated, and the current frame is set as the key frame, if at least one moving pig exists.

Initially, an object detector YOLOv4 is applied to the sample frame, and bounding boxes are acquired. Bounding boxes that have less than a certain confidence value (set to 0.7 in this study) are removed to reduce false positives, and boxes that can be trusted are chosen. The average detection box size $S$ for the sample frame is calculated. Following this, the number of pixels $D$ for which the difference in pixel value is higher than a certain threshold (i.e., $TH_{pixeldiff}$) is calculated for regions in individual detection boxes of the current frame. Subsequently, if the divided value of D to $S \times N$ is higher than a certain threshold ($TH_{keyframe}$), where the N is number of pigs in a pig pen, then the current frame is designated as the key frame. For this study, threshold values that are needed to extract key frames were set to $TH_{pixeldiff} = 1$, $TH_{keyframe} = 1$, and the algorithm to extract the key frame is shown as Algorithm 2.

---

**Algorithm 2.** Key frame extraction algorithm.

---

Input: Current frame CF, Previous key frame PF
Output: Key frame $f_t$
Initialize: Average bounding box size S,
          Pixel difference of CF and PF $D$,
          Number of pigs in pig pen $N$,
        Hyperparameter $TH_{pixeldiff}$, $TH_{keyframe}$
if $D > TH_{pixeldiff}$ **do**
    if $\frac{D}{S \times N} > TH_{keyframe}$ **do**
      return CF
    else **do**
      return 0

---

After extracting the key frames, the detection accuracy is evaluated without box-level annotation. As explained, the maximum number of pigs that can be detected within an enclosed pig pen (i.e., max_pigs) is known. With max_pigs, the accuracy $ACC_{max\_pigs}$ is defined for n test frames. For each test frame i, the number of detection boxes $C_i$ is compared with max_pigs. If they are equal, the frame is considered as the correct frame (i.e., $F_i = 1$). Then, $ACC_{max\_pigs}$ is computed as the ratio of the total number of test frames to the total number of correct frames w.r.t max_pigs.

However, even though the number of detection boxes is equal to max_pigs, there is a possibility of false-positive and/or false-negative errors. As explained, the number of pigs by manual inspection ($GT_i$) for each test frame i can be less than max_pigs because of possible occlusion. Therefore, with $GT_i$, an accuracy $ACC_{manual\_inspection}$ for n test frames is defined. For each test frame i, the number of detection boxes $C_i$ is compared to the value of $GT_i$. If they are equal, the frame is considered as the correct frame (i.e., $M_i = 1$). Then, $ACC_{manual\_inspection}$ is computed as the ratio of the total number of test frames to the total number of correct frames for manual inspection.

Because the cost of manual inspection is still high for a relatively large number of key frames, a key frame with at most one pig in overexposed regions is also defined as a hard frame, and $ACC_{max\_pigs}$ and $ACC_{manual\_inspection}$ to detect pigs are evaluated for difficult frames. In other words, the detection accuracy for hard frames can represent the lower bound of the detection accuracy for key frames. In Section 4, $ACC_{max\_pigs}$ is compared between raw video frames, key frames, and hard frames, and $ACC_{manual\_inspection}$ is compared between key frames and hard frames to evaluate the effectiveness of $ACC_{max\_pigs}$ and the accuracy relationship between key frames and hard frames.

$$ACC_{max\_pigs} = \sum_{i=1}^{n} \frac{F_i}{n} \quad ACC_{max\_pigs} = \sum_{i=1}^{n} \frac{F_i}{n} \tag{1}$$

$$F_i = \begin{cases} 1 & \text{if max\_pigs} = C_i \\ 0 & \text{otherwise} \end{cases} \tag{2}$$

$$ACC_{manual\_inspection} = \sum_{i=1}^{n} \frac{M_i}{n} \tag{3}$$

$$M_i = \begin{cases} 1 & \text{if } GT_i = C_i \\ 0 & \text{otherwise} \end{cases} \tag{4}$$

## 4. Experimental Results

### 4.1. Experimental Setup and Resources for the Experiment

For the purpose of comparison, individual pig detection experiments were conducted in the following environment: Intel Core i5-9400F 2.90 GHz (Intel, Santa Clara, CA, USA), NVIDIA GeForce RTX2080 Ti (NVIDIA, Santa Clara, CA, USA), 32 GB RAM, Ubuntu 16.04.2 LTS (Canonical Ltd., London, UK), and OpenCV 3.4 [36] for image processing.

The experiment was conducted in a 3.2 m tall, 2.0 m wide, and 4.9 m long pigsty at Chungbuk National University, and a low-cost Intel RealSense camera (D435 model, Intel, Santa Clara, CA, USA) [37] was installed on the ceiling to obtain the images. A total of nine pigs (Duroc × Landrace × Yorkshire) were raised in a pig pen, and the average initial body weight of each pig was 92.5 ± 5.9) kg. Color, infrared, and depth images were acquired using a low-cost camera installed on the ceiling, and each image had a resolution of 1280 × 720 at 30 frames per second (fps). Figure 5 shows a pig pen with a camera installed on the ceiling. To exclude the unnecessary region of the pig pen, the region of interest (RoI) was set to 608 × 288.

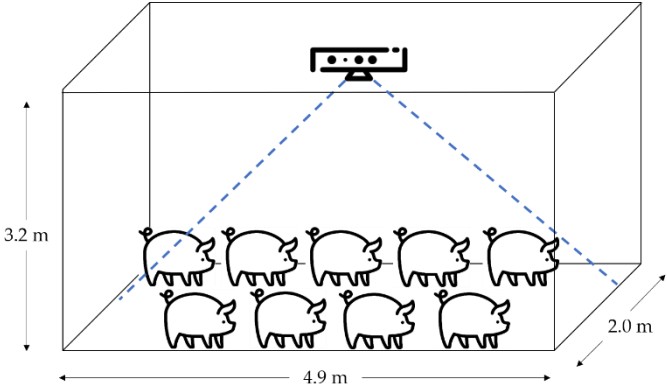

**Figure 5.** Experimental setup with Intel RealSense camera.

From the camera, 2904 training images were acquired, and image preprocessing A and B were applied with the basic image augmentation method (horizontal flip, vertical flip, horizontal/vertical flip). Following this, models A and B were trained (0.0001 for learning rate, 0.0005 for decay, 0.9 for momentum, Mish as the activation function, default anchor parameter, and 6000 for the iterations) to obtain EnsemblePigDet. Then, 216,000 test images were extracted from surveillance videos between 8:30 and 10:30 in the morning, when the pigs were most active. From these, 13,997 key frames and 4193 hard frames were extracted and verified as being exposed to strong sunlight. The reported accuracy was the average of the five-fold cross-validation. The proposed method was implemented based on YOLOv4 [31]. With the COCO data set [32], YOLOv4 exhibited a better tradeoff between speed and accuracy than other detectors, and thus YOLOv4 was selected as the baseline.

*4.2. Evaluation of Detection Performance*

Table 3 shows the experimental results with YOLOv4 as baseline, image augmentation applied with the single model, and ensemble model after applying proposed image preprocessing A and image preprocessing B. Detection accuracy metric $ACC_{max\_pigs}$ symbolizes the accuracy when the number of detected boxes is not nine, though the number of pigs that can be detected within a pen is nine in total (max_pigs = 9). If a falsely detected box and an omitted box are present in a frame, the frame is not considered as an error frame. The frame is considered to be an error frame when the pigs are completely occluded and only eight pigs are visually identified. These are the current limitations of the proposed error frame metric, but the general accuracy of the large number of test data, without visual verification, holds special significance. As shown in Table 3, $ACC_{max\_pigs}$ shows better results for the proposed single model using image preprocessing than the baseline YOLOv4, and the ensemble model that uses the model ensemble method shows the best result. In addition, the miniscule difference between the accuracy of the total frame and the key frame is verified. Using the proposed method, the key frame is only 7% of the total frame, but a conclusion similar to the experimental result on the total frame can be drawn on the key frame, verifying the possibility of effectively reducing the total number of frames. Hence, instead of monitoring all the frames, the extracted key frames from a pig pen can be monitored to reduce the process time and overhead time in terms of the model ensemble

process time using a two-model method. In addition, a movement-based key frame is decided dynamically; however, as only 7% of all frames captured in the morning show pigs with significant movement, the process time in the ensemble model of the keyframe is practically faster than the process time in a single model of the entire frame.

**Table 3.** Comparison of accuracy $\text{ACC}_{\text{max\_pigs}}$ for 216,000 raw video frames and 13,997 key frames with overexposed regions (obtained during 8:30 a.m. to 10:30 a.m.).

| Model | | # Error Frames with max_pigs $(\text{ACC}_{\text{max\_pigs}})$ | |
|---|---|---|---|
| | | 216,000 Raw Video Frames | 13,997 Key Frames |
| Single Model | Baseline YOLOv4 | 43,344 (79.93%) | 3976 (71.59%) |
| | Model A (proposed) | 20,248 (90.63%) | 1619 (88.43%) |
| | Model B (proposed) | 21,092 (90.24%) | 1844 (86.83%) |
| Ensemble Model | EnsemblePigDet (proposed) | 12,244 (94.33%) | 621 (95.56%) |

In addition, experiments were commenced to validate the suppression effect of the decreasing accuracy problem based on the pigs exposed to sunlight, proposed in this study. Table 4 shows the results of the experiments on extracted hard frames that contain pigs exposed to strong sunlight. When compared to Table 4, the overall accuracy decreased in all of the models when exposed to strong sunlight, as shown in Table 4, but the decrease in accuracy is relatively small for the proposed single model and the ensemble model, compared to baseline YOLOv4. Therefore, models trained on data applied with image preprocessing A and image preprocessing B are shown to have a significant effect in the presence of strong sunlight, which can be increased through the model ensemble.

**Table 4.** Comparison of accuracy $\text{ACC}_{\text{max\_pigs}}$ for 13,997 key frames and 4193 hard frames with overexposed regions (obtained during 8:30 a.m. to 10:30 a.m.).

| Model | | # Error Frames with max_pigs $(\text{ACC}_{\text{max\_pigs}})$ | |
|---|---|---|---|
| | | 13,997 Key Frames | 4193 Hard Frames |
| Single Model | Baseline YOLOv4 | 3976 (71.59%) | 1906 (54.54%) |
| | Model A (proposed) | 1619 (88.43%) | 848 (79.78%) |
| | Model B (proposed) | 1844 (86.83%) | 778 (81.45%) |
| Ensemble Model | EnsemblePigDet (proposed) | 621 (95.56%) | 279 (93.34%) |

However, even if the number of detected boxes and max_pigs are the same, the detected boxes have the possibility of being falsely detected boxes. Therefore, validity and accuracy of the results cannot be confirmed with the number of detections alone. Figure 6a represents the case where the number of detection boxes and max_pigs is equivalent but is considered as an error frame because of the falsely detected box. Furthermore, while the maximum number of pigs that could be detected was nine in the video used in this experiment, there were cases where the number of pigs was not nine, due to severe occlusion. Figure 6b shows the case where a frame would be considered as a correct frame as the number of detection boxes was equivalent to the actual number of pigs when visually checked, even though the number of detection boxes and max_pigs was different. Therefore, the actual detection result was checked visually using the extracted key frame and hard frame. In addition, a total of 864,000 images have to be checked visually to acquire $\text{ACC}_{\text{manual\_inspection}}$ for four models, with 216,000 images each, when test frames are visually checked, instead of a key frame or hard frame. Therefore, a visual test is performed on the key frame and hard frame, instead of the total frame. If the $\text{ACC}_{\text{max\_pigs}}$ value and $\text{ACC}_{\text{manual\_inspection}}$ value on the keyframe and hard frame are not significantly

different, it can be inferred that the $ACC_{max\_pigs}$ value and $ACC_{manual\_inspection}$ value on 216,000 total test frames also does not have a significant difference.

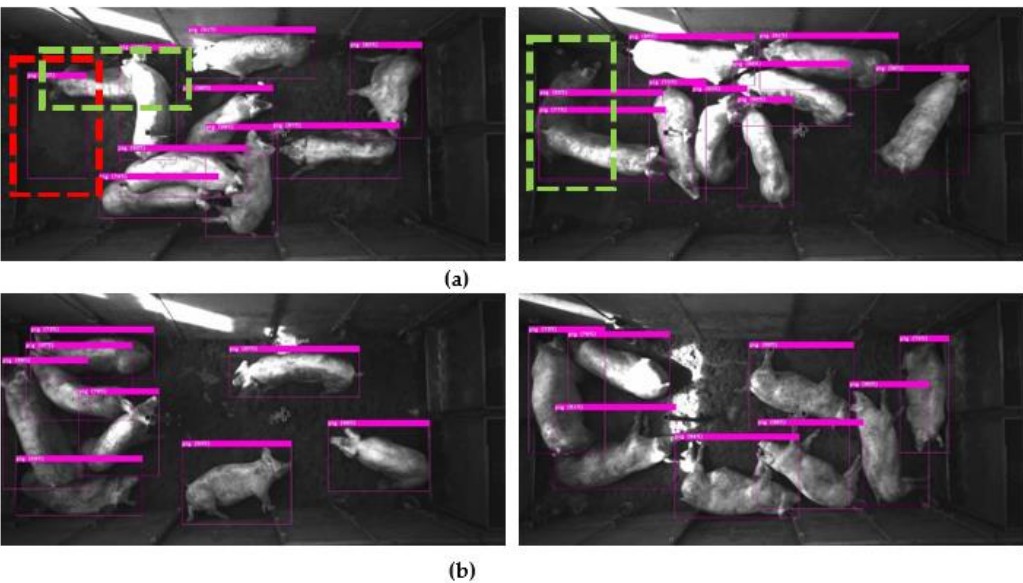

**Figure 6.** Limitations of $ACC_{max\_pigs}$ when max_pigs = 9. (**a**) An "error" frame with one false-positive error and one false-negative error based on ground-truth (although it is regarded as a "correct" frame based on $ACC_{max\_pigs}$); (**b**) a "correct" frame with no false positive/negative error based on ground-truth (although it is regarded as an "error" frame based on $ACC_{max\_pigs}$).

Table 5 shows the result of actual counting with visual checking (hence, $ACC_{manual\_inspection}$), which is considerably similar to the counting result using the number of boxes ($ACC_{max\_pigs}$ shown in Table 5). As explained in Figure 6, there are cases where $ACC_{max\_pigs}$ does not consider a frame to be an error frame when it actually is and deems a frame to be an error frame when it is not. Overall, $ACC_{max\_pigs}$ and $ACC_{manual\_inspection}$ have similar results. Hence, if the test data are too large to waive box annotation for each individual pig or pose difficulties in checking the frames visually, $ACC_{max\_pigs}$ is identified as an actual performance metric to represent the detection accuracy of the deep learning model when determining the number of pigs within a pig pen (max_pigs). In addition, unlike the general detection accuracy metric, which is based on box annotation of individual pigs, AP (average precision)/AR (average recall), $ACC_{max\_pigs}$ and $ACC_{manual\_inspection}$ consider a frame even with one error as an error frame (for example, eight pigs are correctly detected, but if a pig is omitted, then the frame is considered to have an error). Thus, the metrics display fewer numbers overall when compared to AP/AR.

**Table 5.** Comparison of accuracy $ACC_{manual\_inspection}$ for 13,997 key frames and 4193 hard frames with overexposed regions (obtained during 8:30 a.m. to 10:30 a.m.).

| Model | | # Error Frames with Manual Inspection ($ACC_{manual\_inspection}$) | |
|---|---|---|---|
| | | **13,997 Key Frames** | **4193 Hard Frames** |
| Single Model | Baseline YOLOv4 | 3889 (72.21%) | 1866 (55.49%) |
| | Model A (proposed) | 1532 (89.05%) | 808 (80.72%) |
| | Model B (proposed) | 1757 (87.44%) | 738 (82.39%) |
| Ensemble Model | EnsemblePigDet (proposed) | 649 (95.36%) | 302 (92.79%) |

To test the effect on image preprocessing and the model ensemble, a comparison experiment was performed, and Table 6 shows the results. In a single model, as shown in

Table 6, a trained model that included raw input showed higher accuracy than the trained model that applied image preprocessing. In addition, training all of image preprocessing A, image preprocessing B, and raw input together and dividing them into image preprocessing A + raw input and image preprocessing B + raw input showed higher accuracy. In the case of the model ensemble, the ensemble of two models with high accuracy on a single model is shown to have the best accuracy.

**Table 6.** Comparison of accuracy $ACC_{max\_pigs}$ for 13,997 key frames with overexposed regions (obtained during 8:30 a.m. to 10:30 a.m.) with different training data.

| Proposed Models | | # Error Frames with max_pigs ($ACC_{max\_pigs}$) | |
|---|---|---|---|
| | | Test Data with Preprocessing A | Test Data with Preprocessing B |
| Single Model | Training Data with Preprocessing A | 4111 (70.06%) | 13,997 (0.00%) |
| | Training data with Preprocessing B | 6391 (54.34%) | 3287 (76.51%) |
| | Training Data with Preprocessing A and Preprocessing B | 3741 (73.27%) | 4198 (70.00%) |
| | Training Data with Preprocessing A and Raw Input | 1619 (88.43%) Model A | 10,743 (23.24%) |
| | Training Data with Preprocessing B and Raw Input | 3881 (72.27%) | 1844 (86.83%) Model B |
| | Training Data with Preprocessing A and Preprocessing B and Raw Input | 3185 (77.24%) | 3891 (72.20%) |
| Ensemble Model | Training Data with Preprocessing A + Preprocessing B | 2498 (82.15%) | |
| | Training Data with Preprocessing A and Raw Input + Preprocessing B and Raw Input | 621 (95.56%) EnsemblePigDet | |

Tables 7 and 8 show the combination results of Confidence1 and 2 and Threshold1 and 2 for 13,997 key frames. Table 7 shows the result of setting Confidence1 and Threshold1 conservatively and Confidence2 and Threshold2 aggressively. As the Confidence1 and 2 values increased, accuracy showed a decreasing trend. In addition, unrelated to the values of Confidence1 and 2, the accuracy tended to decrease with the increase of the Threshold1 value. Table 8 shows the results of setting Confidence1 and Threshold1 aggressively and Confidence2 and Threshold2 conservatively, exhibiting similar results to Table 7, where the accuracy was shown to decrease as Confidence1 value increased. Nevertheless, unrelated to the confidence value, the accuracy increased when Threshold1 and 2 values increased. The overall accuracy was shown to increase when Confidence1 and Threshold1 were set aggressively and Confidence2 and Threshold2 were set conservatively. When comparing the ensemble model to the single model in Tables 7 and 8, the accuracy was shown to increase. Therefore, based on the combination of confidence and threshold, the ensemble model displays less accuracy than a single model.

**Table 7.** Comparison of accuracy $ACC_{max\_pigs}$ for 13,997 key frames with overexposed regions (obtained during 8:30 a.m. to 10:30 a.m.) with different ensemble parameters (case of Confidence1 and Threshold1 = conservative, Confidence2 and Threshold2 = aggressive).

| Proposed Ensemble Models | | # Error Frames with max_pigs ($ACC_{max\_pigs}$) |
|---|---|---|
| Confidence1 = 0.3 and Confidence2 = 0.5 | Threshold1 = 0.5 and Threshold2 = 0.3 | 956 (93.17%) |
| | Threshold1 = 0.7 and Threshold2 = 0.3 | 1162 (91.70%) |
| | Threshold1 = 0.7 and Threshold2 = 0.5 | 1077 (92.31%) |
| Confidence1 = 0.3 and Confidence2 = 0.7 | Threshold1 = 0.5 and Threshold2 = 0.3 | 1173 (91.62%) |
| | Threshold1 = 0.7 and Threshold2 = 0.3 | 3092 (77.91%) |
| | Threshold1 = 0.7 and Threshold2 = 0.5 | 3090 (77.92%) |
| Confidence1 = 0.5 and Confidence2 = 0.7 | Threshold1 = 0.5 and Threshold2 = 0.3 | 1753 (87.48%) |
| | Threshold1 = 0.7 and Threshold2 = 0.3 | 2283 (83.69%) |
| | Threshold1 = 0.7 and Threshold2 = 0.5 | 2271 (83.78%) |

**Table 8.** Comparison of accuracy $ACC_{max\_pigs}$ for 13,997 key frames with overexposed regions (obtained during 8:30 a.m. to 10:30 a.m.) with different ensemble parameters (case of Confidence1 and Threshold1 = aggressive, Confidence2 and Threshold2 = conservative).

| Proposed Ensemble Models | | # Error Frames with max_pigs ($ACC_{max\_pigs}$) |
|---|---|---|
| Confidence1 = 0.5 and Confidence2 = 0.3 | Threshold1 = 0.3 and Threshold2 = 0.5 | 832 (94.06%) |
| | Threshold1 = 0.3 and Threshold2 = 0.7 | 637 (95.45%) |
| | Threshold1 = 0.5 and Threshold2 = 0.7 | 621 (95.56%) EnsemblePigDet |
| Confidence1 = 0.7 and Confidence2 = 0.3 | Threshold1 = 0.3 and Threshold2 = 0.5 | 916 (93.46%) |
| | Threshold1 = 0.3 and Threshold2 = 0.7 | 678 (95.16%) |
| | Threshold1 = 0.5 and Threshold2 = 0.7 | 673 (95.19%) |
| Confidence1 = 0.7 and Confidence2 = 0.5 | Threshold1 = 0.3 and Threshold2 = 0.5 | 1312 (90.63%) |
| | Threshold1 = 0.3 and Threshold2 = 0.7 | 1361 (90.28%) |
| | Threshold1 = 0.5 and Threshold2 = 0.7 | 1353 (90.33%) |

*4.3. Discussion*

Figure 7 shows the result of solving the failed detection cases for each model using the model ensemble. In the case of Figure 7a, while false negatives were created that resulted in missing pig detection in model B, all of the pigs within a frame detected through the model ensemble are shown. In the case of Figure 7b, while false positives were created that resulted in the detection of background as pig in model A, all of the pigs detected through the model ensemble that merged with falsely detected boxes are shown. In addition, Figure 7c shows false negatives and false positives created in model B, which were merged using a model ensemble that correctly detected all pigs within a frame. Hence, even if falsely detected boxes were created from each model, enhancing the detection result of each model and further merging the result using the model ensemble was possible.

Although many errors with single models could be solved by the ensemble model, some errors still remain. Figure 8 shows the result of not being able to detect all pigs within a frame, even after applying the model ensemble. In the case of Figure 8a, while false negatives occurred in models A and B, detection of all pigs within a frame failed even after merging the result using the model ensemble because of false negatives occurring for the same pig. In addition, as shown in Figure 8b, even though false negatives of model A and false positives of model B were created, the detection boxes could not be merged. This was because the false positives of model B were considered as pigs instead of the false negatives of model A, even after applying the model ensemble, due to false positives with high confidence scores. To summarize, the detection result might not merge correctly, even after applying the model ensemble to the following cases: the same pigs not being detected,

highest confidence score of false positive, and the previous case with the false negative and false positive occurring at the same time.

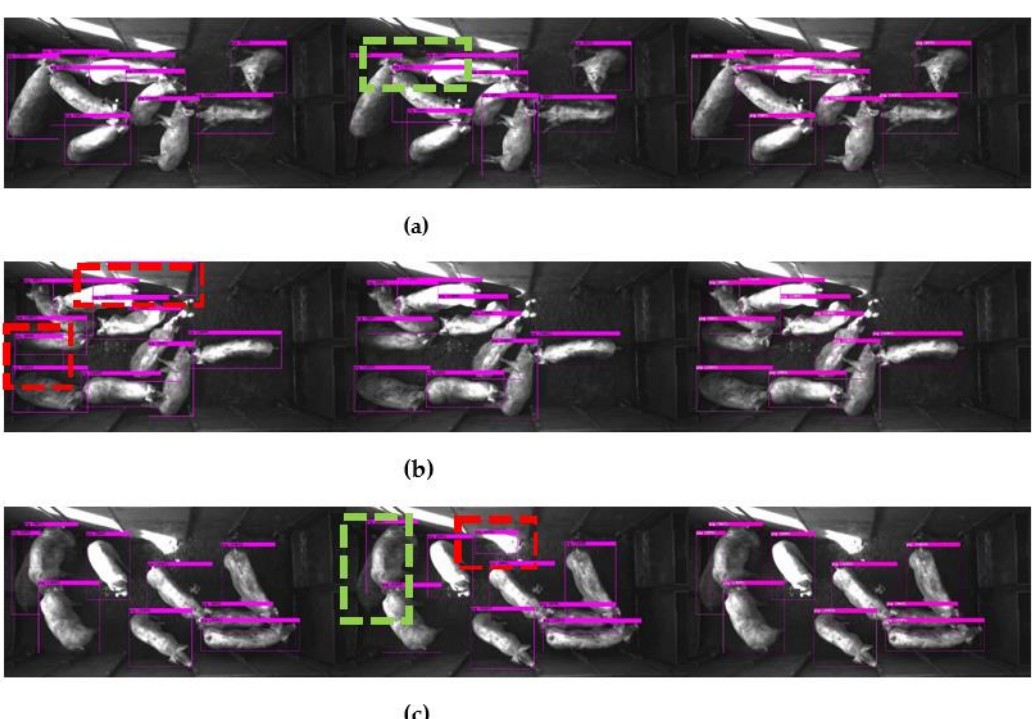

**Figure 7.** Results of model ensemble (**left**: model A, **center**: model B, **right**: EnsemblePigDet). (**a**) Cases of missing pigs in single model; (**b**) cases of false pigs in single model; (**c**) cases of missing and false pigs in single model.

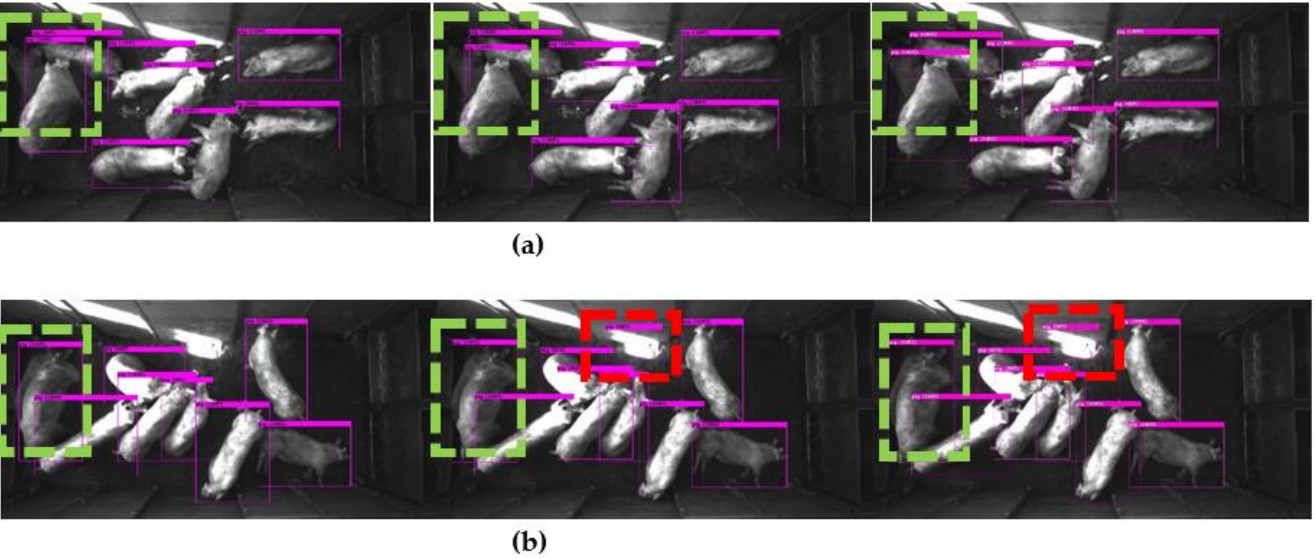

**Figure 8.** Failure cases (**left**: model A, **center**: model B, **right**: EnsemblePigDet). (**a**) Cases of missing pigs in single model. (**b**) Cases of missing and false pigs in ensemble model.

Although the proposed method could improve the accuracy of baseline YOLOv4 significantly for pig detection from unseen overexposed regions, the limitations of this study are as follows:

- While research on unseen data that include strong sunlight in the same farm was performed, the development of a more robust model (through semi-supervised or

self-supervised learning) using unseen data from other farms might be necessary in future research. In addition, the remaining errors with each key frame could be solved by exploiting the temporal information among the key frames; thus, this issue could be addressed in future research.

- As shown in Table 9, the accuracy improvement of EnsemblePigDet was strongly dependent on the accuracy of the baseline model used. Because all the 13,997 key frames were considered as difficult images due to overexposed regions, $ACC_{max\_pigs}$ of the light-weight model was significantly degraded. Note that EmbeddedPigDet [26] modified TinyYOLOv2 for embedded board implementations, and 13,962 key frames (from 13,997 total key frames) typically produced one or two errors (i.e., missing and/or false pig errors) in each keyframe with EmbeddedPigDet. That is, EmbeddedPigDet targeted for embedded board implementations cannot be used for a hard scenario including strong sunlight, and the accuracy improvement of EnsemblePigDet based on EmbeddedPigDet was limited. Ensemble techniques for light-weight baseline models need to be studied further.

- In this study, even though fast and accurate YOLOv4 was applied, the execution time of the ensemble model (i.e., the total time on a PC for processing one input image was 29.84 ms, with 5.22 ms for two preprocessing executions, 24.24 ms for two YOLOv4 executions, and 0.38 msec for one postprocessing execution) was slower than those of single models. However, if detection was applied to key frames that captured movements through pig pen monitoring using the proposed method, an average of 20-fold (extracting 13,997 key frames of 216,000 frames using the key frame extraction method) reduction of computation complexity was verified. Therefore, with the RTX2080 Ti GPU, the detection speed of the video composed of key frames was 17 times faster than that of the raw video (i.e., 7 min were required for processing 13,997 key frames obtained from the two-hour raw video); thus, the proposed method with key frames could be executed in real time even on an embedded board.

**Table 9.** Comparison of performance for 13,997 key frames with overexposed regions (obtained during 8:30 a.m. to 10:30 a.m.).

| Model | | # Error Frames with max_pigs | Execution Time | |
|---|---|---|---|---|
| | | ($ACC_{max\_pigs}$) | PC | Embedded Board |
| | | | (RTX2080Ti) | (Xavier NX [38]) |
| YOLOv4 [31] | Baseline | 3976 (71.59%) | 168 s ≈3 min | 2661 s ≈44 min |
| | EnsemblePigDet (proposed) | 621 (95.56%) | 417 s ≈7 min | 5427 s ≈90 min |
| TinyYOLOv4 [31] | Baseline | 4886 (65.09%) | 120 s ≈2 min | 379 s ≈6 min |
| | EnsemblePigDet (proposed) † | 3652 (73.90%) | 231 s ≈4 min | 762 s ≈13 min |
| Embedded PigDet [26] * | Baseline | 13,962 (0.25%) | 42 s ≈1 min | 265 s ≈5 min |
| | EnsemblePigDet (proposed) † | 13,469 (3.77%) | 94 s ≈2 min | 532 s ≈9 min |

†: For the purpose of comparison, we used the parameters determined with baseline YOLOv4. *: For the purpose of comparison, we reimplemented EmbeddedPigDet [26] for grayscale images rather than composite images.

## 5. Conclusions

Automated pig monitoring is important for smart pig farms; thus, several deep-learning-based pig monitoring techniques have been proposed recently. In applying automated pig monitoring techniques to real pig farms, however, practical issues such as detecting pigs from overexposed regions, caused by strong sunlight through a window, should be considered. Another practical issue in applying deep-learning-based techniques to a specific pig monitoring application is the annotation cost for pig data.

In this study, a method for managing these two practical issues is proposed. Using annotated data obtained from training images without such overexposed regions, augmented data were generated to reduce the effect of the overexposed condition. Then, YOLOv4 was trained with the annotated as well as augmented data. The test results from two YOLOv4 models were combined in a bounding box level to further improve the detection accuracy. Finally, accuracy metrics were proposed for pig detection in a closed pig pen to evaluate its accuracy with no box-level annotation.

The experimental results with 216,000 "large-scale unseen" test data with overexposed regions in the same pig pen showed that the proposed ensemble method could significantly improve the detection accuracy to 94% from 79% of the baseline of YOLOv4. In addition, the accuracy for 216,000 raw video frames was consistent with that of 13,997 key frames, and the accuracy for 4193 hard frames could provide the lower bound of the accuracy for 13,997 key frames. One limitation of the proposed method is the increased execution time. Although the proposed method with key frame extraction could be executed in real time (i.e., 2 h test data could be processed without any delay) even on an embedded board, a method for reducing the increased execution time needs to be studied further.

**Author Contributions:** S.L. and Y.C. conceptualized and designed the experiments; H.A., S.S. and H.K. designed and implemented the detection system; Y.C. and D.P. validated the proposed method; H.A. and S.S. wrote the paper. All authors have read and agreed to the published version of the manuscript.

**Funding:** This research was supported by the Basic Science Research Program through the NRF, funded by the MOE (2018R1D1A1A09081924).

**Institutional Review Board Statement:** Not applicable.

**Informed Consent Statement:** Not applicable.

**Data Availability Statement:** Not applicable.

**Conflicts of Interest:** The authors declare no conflict of interest.

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
