# Peer review of "EnsemblePigDet: Ensemble Deep Learning for Accurate Pig Detection"

_applsci, doi:10.3390/app11125577_

Round 1
Reviewer 1 Report
The aim of this study is to identify the number of pigs in a pig pen using Yolov4. This article addressed an interesting topic that is relevant to practice. However, similar study can be found in same journal, and the novelty of this article is unclear. Actually, this study is to propose an image preprocessing to reduce the annotation cost and detect pig more correctly without box-level annotation. Also, this article more like a technical report. Some of my specific comments are listed below.
- Title; title should be revised to highlight the contribution of this article.
- Line 100-101; actually, pig detection with large-scale test data have reported in similar studies, such as “EmbeddedPigDet—Fast and Accurate Pig Detection for Embedded Board Implementations”.
- Line 148-150, and Lin 72 and 164; lots of sentences with the same meaning were repeated. This article needs to be reorganized. In addition, lots of same figures were repeated in similar studies.
- Line 305-306; the author should clarify how to know and identify key frames.
- Line 373-374; does this research use color and depth images for the experiment?
- Line 572; the device with RTX2080 usually is rarely used in practical detection. In other words, the author should prove that the proposed method can be used practical pig detection from overexposed regions.
- Similar studies “EmbeddedPigDet—Fast and Accurate Pig Detection for Embedded Board Implementations” and “Automated Individual Pig Localisation, Tracking and Behaviour Metric Extraction Using Deep Learning”, could be found, In other words, the novelty of this article should be clearly stated.
Reviewer 2 Report
paper is interesting.. covering real investigation on important topic.
results are clear and promissing!
unfortunately i recomend to improve introduction, methodology, references and mainly to add discussion!
references need to be places into the text in whole article. they need to be used one by one.. not at groups like at introduction...
mostly from journal sources.. Q1/Q2 are the most suggested
in the discussion you need to refer to other journal sources and compare your results with other work.. to provide more reliable discussion..
in conclusion please compare contribution to other relevant authors.. what are PROS and CONS of your solution..
same need to be in asbtract! to atract readers...
after incorporating these MINOR changes, article can be accepted
Reviewer 3 Report
The paper proposed a method to detect pigs from overexposed regions and apply deep learning to support pig monitoring application. The idea is innovative and the work is important and within the scope. Here are some suggestions for improvements:
1. One of the biggest issues of this work is that pig monitoring is not equal to just label them out in the images. There are little chances for the pigs to disappear or their numbers will change in one of farms. What is the real point of bounding boxing them out from the surveillance images? I think something like detecting the health status or the behaviors (sleeping, eating, walking, standing, etc) of each pig will be much more meaningful to realize "smart" pig farms than just detecting their existence in those images. The annotation should focus on labeling the status of pigs instead of just drawing rectangles.
2. The sunlight through window problem is a common issue in deep learning models. It is really a limitation from the used image bands. If some other bands like Near Infrared are added into the inputs, the problem might be gone.
3. As a deep learning research, the generalization of the trained models is not discussed. What if people want to reuse the model in another pig pen where the surveillance cameras are installed with a different angle instead of vertically on the top? Will the trained model be reusable or people need do the labeling-training again?
4. As mentioned in the paper, the practical challenge of using deep learning is the cost of annotation. It takes 5 minutes to annotate an image. How many samples are considered adequate to train a working model and how fast can it be to deploy a deep learning model from scratch? Elaborate some details on the cost management of this approach and it would help people to estimate on adopting the approach.
Round 2
Reviewer 1 Report
Similar studies “EmbeddedPigDet—Fast and Accurate Pig Detection for Embedded Board Implementations” and “Automated Individual Pig Localisation, Tracking and Behaviour Metric Extraction Using Deep Learning”, could be found. The author should clearly state where this article is different from those articles
Reviewer 3 Report
The authors addressed all my concerns.
Author Response
Reviewer #3:
Comments and Suggestions for Authors
The authors addressed all my concerns.
â–º Response: Thank you very much for your helpful comment. If there is no problem with our paper, please sign the review report.
